# Machine Learning-Based Segmentation of the Thoracic Aorta with Congenital Valve Disease Using MRI

**DOI:** 10.3390/bioengineering10101216

**Published:** 2023-10-18

**Authors:** Elias Sundström, Marco Laudato

**Affiliations:** 1Department of Engineering Mechanics, FLOW Research Center, KTH Royal Institute of Technology, Teknikringen 8, 10044 Stockholm, Sweden; 2Department of Engineering Mechanics, The Marcus Wallenberg Laboratory for Sound and Vibration Research, KTH Royal Institute of Technology, Teknikringen 8, 10044 Stockholm, Sweden

**Keywords:** machine learning segmentation, 4D-PCMRI, aortic valve disease

## Abstract

Subjects with bicuspid aortic valves (BAV) are at risk of developing valve dysfunction and need regular clinical imaging surveillance. Management of BAV involves manual and time-consuming segmentation of the aorta for assessing left ventricular function, jet velocity, gradient, shear stress, and valve area with aortic valve stenosis. This paper aims to employ machine learning-based (ML) segmentation as a potential for improved BAV assessment and reducing manual bias. The focus is on quantifying the relationship between valve morphology and vortical structures, and analyzing how valve morphology influences the aorta’s susceptibility to shear stress that may lead to valve incompetence. The ML-based segmentation that is employed is trained on whole-body Computed Tomography (CT). Magnetic Resonance Imaging (MRI) is acquired from six subjects, three with tricuspid aortic valves (TAV) and three functionally BAV, with right–left leaflet fusion. These are used for segmentation of the cardiovascular system and delineation of four-dimensional phase-contrast magnetic resonance imaging (4D-PCMRI) for quantification of vortical structures and wall shear stress. The ML-based segmentation model exhibits a high Dice score (0.86) for the heart organ, indicating a robust segmentation. However, the Dice score for the thoracic aorta is comparatively poor (0.72). It is found that wall shear stress is predominantly symmetric in TAVs. BAVs exhibit highly asymmetric wall shear stress, with the region opposite the fused coronary leaflets experiencing elevated tangential wall shear stress. This is due to the higher tangential velocity explained by helical flow, proximally of the sinutubal junction of the ascending aorta. ML-based segmentation not only reduces the runtime of assessing the hemodynamic effectiveness, but also identifies the significance of the tangential wall shear stress in addition to the axial wall shear stress that may lead to the progression of valve incompetence in BAVs, which could guide potential adjustments in surgical interventions.

## 1. Introduction

The aorta is the main artery in the human body, with the life-sustaining role of distributing oxygenated blood to all parts of the body via systemic circulation. With its complex morphology, it has been extensively researched to explore the space of parameters that can impact the biomechanical function of the aorta [1]. The bicuspid aortic valve (BAV) is a congenital aortic valvar disease present in 1–2 percent of the population. It is characterized by a fusion of the right and left coronary leaflets, but can also show other geometric variations on the raphe length, interleaflet triangle, and the rotational position of the aortic valve. Such rotational position has been hypothesized to influence wall shear stress (WSS) and helicity of blood flow in the ascending aorta [2,3,4,5]. This is believed not only to affect the valve competency, but also increase the risk for aortic dilation [6]. Additionally, the calcification of aortic valvar leaflets can lead to complications such as aortic regurgitation, increased shear stress, and pressure loss [7].

Meierhofer et al. [8] utilized three-dimensional time-resolved phase-contrast magnetic resonance imaging (4D-PCMRI) to compare blood flow patterns between individuals with BAV and those with normal tricuspid aortic valves (TAV). They found that non-stenotic BAVs exhibit higher tangential shear stress but lower axial shear stress compared to TAV, which has been confirmed through in vitro studies [9]. Furthermore, qualitative analysis has shown that vortical flow structures, which potentially contribute to aortic dilation, are more prevalent in BAVs. However, the scales and strength of these vortical structures were not quantified [8]. Similarly, Dux-Santoy et al. [10] quantified higher WSS magnitude in non-stenotic BAV compared to healthy volunteers but found no correlation with the pathogenesis of aortic dilation. In two related 4D-PCMRI studies, it was observed that the jet angle emanating from the valve differs in TAV and BAV subjects [11,12]. This variation was also reported in the assessment of 4D-PCMRI data by Barker [13], who observed a correlation of the jet impingement on the aortic wall in functional BAVs to coincide with the opposite side of the fused leaflets. The use of 4D-PCMRI has also increased the understanding of aortic stiffness associated with aortic disease, degenerative aneurysms, and chronic dissections [14,15]. Hope et al. [16] investigated 4D-PCMRI data and hypothesized that counter-rotating helices in the ascending aorta play a role in reducing velocity fluctuations and lessening the wall shear stress. MRI has also been used in retrospective studies of non-atherosclerotic aortic arch pathologies (NA-AAPs), including conditions like bicuspid aortic valve, inflammatory diseases, and heritable connective tissue disorders. Certain aortic arch variations, like bovine arch and vascular rings, can lead to aortic wall stiffening, promoting atherosclerosis growth and aneurysm formation, highlighting arterial stiffness as a significant risk factor for cardiovascular outcomes [17].

The analysis of 4D-PCMRI faces challenges due to its relatively long lead times, requiring manual segmentation and the process of identification of anatomical landmarks for localizing flow characteristics. To address these limitations, machine learning (ML) algorithms have been employed to fully automate the cardiac imaging workflow [18,19]. Previous studies have used ML algorithms to identify vascular anatomical landmarks in various imaging modalities. It has also been used in emergency clinical scenarios to facilitate accurate diagnosis of different thoracic aortic pathologies [20]. Notably, the 3D U-net convolutional neural network (see next section for details) has shown promise in aortic segmentation using PCMRI [21]. However, due to the data requirements of ML algorithms and the scarcity of large 4D-PCMRI datasets with consistent annotations, limited research exists on the application of ML algorithms for analyzing 4D-PCMRI data.

The study aims to test ML-based segmentation using MRI data to analyze vortical structures and wall shear stress in subjects with BAV and contrast it against normal TAV. With the use of ML-based segmentation, this study aims to provide a physics-based understanding of the biomechanical characteristics of the aorta and provide valuable insights for clinical practice.

## 2. Method

### 2.1. MRI Acquisition

The present study used cardiac MRI data sets acquired at Cincinnati Children’s Hospital Medical Center (CCHMC). All demographic information was anonymized (i.e., age, sex, etc.) and CCHMC Institutional Review Board deemed the study to be exempt from any ethical inquiries [2,3,4]. These data sets were acquired from six subjects, three individuals with non-stenotic tricuspid aortic valves (TAV) and trisinuate aortic roots, as well as three individuals with non-stenotic functionally bicuspid aortic valves (BAV) featuring right–left coronary leaflet fusion (specifically type 1 fusion according to Sievers et al. [22]).

All MRI scans were performed using a 1.5 Tesla clinical MRI scanner (Ingenia, Philips Healthcare; Best, Netherlands) using a phased-array coil. The study of the protocol for the long-axis sagittal stack 4D phase-contrast magnetic resonance imaging (4D-PCMRI), the short-axis aortic root cine stack, and the aortic root phase-contrast velocity sequence was carried out, in addition to the noncontrast 3D mDixon angiogram for analysis. The short-axis aortic root cine stack was obtained using a steady-state free precession pulse sequence with specific parameters: a repetition time of 7.8 ms, an echo time of 4.7 ms, a flip angle of 15°, and sequential 2 mm slices without an interslice gap. For the short-axis aortic root phase-contrast velocity encoded sequence, a gradient echo sequence was employed with a repetition time of 4.3 ms, an echo time of 2.7 ms, a flip angle of 12°, and a slice thickness of 6 mm. The encoding velocity for this sequence was set at 1.5 m/s. Each cardiac cycle was represented by 30 phases for both sequences, resulting in a mean temporal resolution of 30–40 ms. The non-contrast coronal 3D mDixon angiogram was acquired using a repetition time of 5.3 ms, a flip angle of 15°, and 1 mm slices with no interslice gap. The 4D-PCMRI sagittal stack was obtained using a velocity encoding of 2 m/s, s repetition time of 3.5 ms, an echo time of 1.9 ms, and a flip angle of 8° [2,3,4].

### 2.2. ML-Based Segmentation

Semantic segmentation is one of the oldest problems in computer vision [23]. It is defined as the ability to label every pixel of an image, even when the object under analysis is completely unknown [24]. In this perspective, segmentation represents a more complex task than object recognition. The latter, indeed, is limited to classifying objects in an image within a set of a-priori specified labels. Segmentation, on the other hand, is a more general problem as it requires the computer to identify and isolate unknown objects.

It is possible to approximately classify the many instances of ML implementations for semantic segmentation problems in three main groups: weakly supervised methods [25], region-based semantic segmentation [26], and fully convolutional network (FCN)-based segmentation [27]. While weakly supervised methods have the advantage of not requiring any labeling of the training data set, they show poor performances in terms of object localization [28]. The other methods can be framed as supervised learning implementation and are based on Convolutional Neural Networks (CNN). One relevant example is U-Net [29]. Its architecture is based on FCN, but is characterized by the presence of multiple up-sampling layers. Essentially, the first half (contractive path) of a U-Net implementation can be seen as a classical contracting CNN, while the second half (expansive path) is symmetrically growing again using up-sampling operators (see Figure 1).

A particularly successful U-Net based implementation specialized in biomedical images is the so-called nnU-Net [30]. Its main feature is the possibility to automatically configure all the hyperparameters of the network by modeling them in terms of fixed parameters, interdependent heuristic rules, and empirical decisions. Such configuration does not require any manual intervention and it can provide a highly accurate segmentation on the 23 public data sets usually employed in biomedical segmentation competitions [31]. A particularly convenient implementation available is called TotalSegmentator [18], which can segment 104 anatomical structures in the human body with a Dice similarity coefficient score of 0.943. The training data set consists of 1368 CT images, manually labeled. The architecture follows the original encoder–decoder nnU-Net scheme (see Figure 1) with the following minor modifications. The activation function employed for the network is the leaky RELUs which has a negative slope of 0.01. Moreover, instance normalization in place of standard normalization is employed, as the batch size is relatively small. Strided convolution is implemented for the down-sampling, whereas the up-sampling is obtained via convolution transposed. The training runs over 1000 epochs, where one epoch includes 250 randomly chosen mini-batches. The algorithm employed in the training is a stochastic gradient descent with an initial learning rate of 0.01, which is then dynamically modified during the training. The loss function is cross-entropy summed with the Dice score. The images are normalized, re-sampled, and then processed by the neural network using a sliding window. The re-sampling, in particular, is a crucial step, as often in the medical domain the information is arranged on nonhomogeneous grids. The information is therefore arranged on a homogeneous grid using 3-spline-based interpolation. A fine feature of nnU-Net is that it automatically adapts its topology to the GPU memory budget. In particular, the algorithm seeks the largest sustainable patch size, which is in turn connected to more contextual information. Finally, the default convolutional kernel size is 3×3×3. However, as medical data often show a different resolution along one axis, the network is able to automatically set the kernel dimension in that direction to 1.

The main advantage of TotalSegmentator is that it can segment a wide range of clinical data (also on pathological cases) with superior performances concerning other publicly available algorithms [32,33,34]. Clearly, the Dice score alone is not enough to provide a full measure of the accuracy of the segmentation. Typical failure cases are the missing small parts of anatomical structures and the mixing of neighboring parts. However, the main limitation is that the training data set consists only of CT data. Consequently, the performance of the segmentation for any other kind of clinical data needs to be investigated. One of the goals of this work is to test the TotalSegmentator’s performance on MRI data to study the cardiovascular assessment of vortical structures and wall shear stress connected to valve incompetence. As discussed in detail in the next section, although the Dice score is comparatively worse (0.8) on MRI data, such an ML-based segmentation is able to reduce the runtime of the cardiac assessment under analysis and allow for the subsequent flow analysis.

## 3. Result

### 3.1. Segmentation Evaluation

The ML-based segmentation algorithm identified most of the larger cardiovascular structures: left atrium, left ventricle, myocardium, right atrium, right ventricle, pulmonary artery, and aorta; see Figure 2. The axial cut through the major chambers of the heart shows a good qualitative agreement with the background MRI. The segmented aorta (light green) was broadly underestimated compared to the manual segmentation (olive green). Details of the aortic root, the leaflets, coronary arteries, as well as the head and neck arteries, i.e., brachiocephalic artery, left common carotid artery and left subclavian artery, were not identified in the dataset. However, a small fraction towards the descending aorta was identified in both TAV and BAV cases. The segmented cross-sectional areas in Figure 2 (bottom row) are quantified in Table 1 compared to the ground truth (in parentheses) and including the Dice score (in square brackets). The Dice score of the left ventricle segmentation was 0.86 ± 0.06 for the TAV cases and 0.89 ± 0.06 for the BAV cases. The Dice score of the segmented aorta, i.e., of the region that was identified, was 0.72 ± 0.12 for the TAV cases and 0.82 ± 0.06 for the BAV cases.

### 3.2. Segmentation Runtime

Table 2 provides a summary of the runtime, RAM (random access memory), and GPU (graphics processing unit) memory requirements for the MRI resolution analysis of both TAV and BAV cases. Both cases cover the thorax and abdomen, with a voxel size of 320 × 320 × 100 in the TAV and 400 × 400 × 100 voxels in the BAV. The runtime and RAM and GPU memory requirements were monitored on a Linux workstation with an Intel Core i9 5.2 GHz CPU and an Nvidia GeForce GTX 1050 Ti. Overall, the runtime of the whole heart segmentation clocked in at about 2 min, which is significantly faster than the manual segmentation that takes in the order of a day (c.f., olive green colored aorta in Figure 1).

### 3.3. Flow Rate, Pulse Wave Velocity, and Arterial Distensibility

Figure 3a compares the measured flow rates between the TAV and BAV cases. Notably, the BAV case exhibits a higher peak flow rate compared to the TAV case. When considering the normalized timescale, the TAV case demonstrates a larger fraction between the systolic and diastolic phases than the BAV case. Both cases exhibit minimal regurgitant fraction, and the net flow fraction between the descending and ascending flow is larger in the BAV case. Table 3 provides an overview of the cardiac output.

The calculation of pulse wave velocity (PWV) involves two crucial parameters: the Aortic Length measurement and the time interval between the upslopes of the flow curves. The time interval is determined by measuring the temporal distance from the point where the tangent of the aortic ascending flow curve reaches zero to the point where the tangent of the descending aorta curve also reaches zero. The calculated PWV values are 3.2 m/s for the TAV case and 3.3 m/s for the BAV case, indicating similar arterial stiffness. The distensibility was assessed using the relation $PWV=\sqrt{1/\rho D}$. The change in area (ΔA/A) is measured in the ascending aorta between peak systolic and the end of diastole, as shown in Figure 3b. The pressure drop is similar for both cases and is determined by the fraction of the area change over the distensibility.

Figure 3c shows the velocity in the ascending aorta, where the error bar represents the standard deviation over time. Both the TAV and BAV cases indicate similar velocity at peak systole (around 60 cm/s), which is due to the larger cross-sectional area in the TAV case. Although there is a slightly larger velocity in the BAV case at peak systole that results in a higher peak kinetic energy compared to the TAV case; see Figure 3d.

### 3.4. Vortical Structures

Figure 4 provides an overview of the blood flow through the left atrium (LA), left ventricle (LV), and aorta during different stages of the cardiac cycle for both the TAV and BAV cases. In the early systole (at approximately t/T=0.05), the aortic valve opens, initiating the ejection of blood from the LV. At the narrower section of the valve, the local flow velocity starts to accelerate, and the streamlines indicate a smooth and streamlined flow in both the TAV and BAV cases. The flow through the valve forms a jet, with the BAV case exhibiting a higher peak velocity that impacts more on the convex tissue wall near the sinotubular junction compared to the TAV case.

Around the time of peak systole (between t/T values of 0.1 and 0.3), the velocities in the descending aorta increase, which corresponds to the pulse wave velocity assessment and the time delay for the pulse wave to propagate from the ascending to the descending aorta. In the TAV case, the streamlines remain relatively aligned with the aorta. However, in the BAV case, some streamlines curl around a strong counterclockwise rotating vortex towards the concave side of the ascending aorta. This vortex qualitatively grows in size, and during post-peak systole (between t/T values of 0.2 and 0.3), it occupies a significant portion of the region proximal to the aortic root and interacts with upper arterial branches, i.e., the brachiocephalic artery and the left common carotid artery. Beyond the head and neck vessels, the flow aligns with the proximal thoracic descending aorta.

As the systolic phase concludes, the valve closes, and the blood flow diminishes, coinciding with a decrease in aortic pressure during diastole. As time progresses to approximately t/T=0.7, in mid-diastole, the mitral valve opens, allowing blood flow from the left atrium to the left ventricle. This is accompanied by an increased flow velocity, facilitating the filling of the LV with fresh blood in preparation for another systole.

Figure 5 and Figure 6 provide additional information on the blood flow through a short axis view, complementing the streamlines shown in the previous Figure 4. The former illustrates the velocity magnitude, while the latter displays the axial vorticity. These figures depict the same time instances of the cardiac cycle, starting from systole when the valve opens.

Around peak systole (between t/T values of 0.1 and 0.3), both the TAV and BAV cases exhibit vortical flow patterns. In the TAV case, the streamlines are directed towards the convex side of the aorta, aligning with the curvature of the ascending aorta. Simultaneously, two secondary Dean-like counter-rotating vortices form along the perpendicular axis to the curvature. However, these vortices are not perfectly symmetric, with slightly higher vorticity observed in the vortex towards the negative y-axis, corresponding to the left cusp of the aortic valve.

In the BAV case, the flow generates a robust swirling motion characterized by positive axial vorticity, indicating counter-clockwise rotation. The streamlines near the center of the vortex converge between t/T values of 0.1 and 0.2, indicating an increase in vortex strength. Beyond peak systole, the kinetic energy of the flow begins to diminish, resulting in reduced magnitudes of both velocity and axial vorticity.

### 3.5. Wall Shear Stress

The blood flow depicted in Figure 4, Figure 5 and Figure 6 is further analyzed along a horizontal profile indicated by a white dashed line in Figure 5. In the TAV case, the axial velocity exhibits small magnitudes at the beginning of systole, progressively increasing towards peak systole and forming a symmetric top-hat profile, see Figure 7a. This behavior suggests the development of a boundary layer with a steeper gradient near the endothelium. Consequently, the axial shear stress component (Figure 7c) shows lower magnitudes at the center and gradually increases towards the endothelial wall. This result is consistent for all considered cases, which can be seen in the evaluation of the mean and standard deviation of the velocity and shear stresses at peak systole; see Figure 8.

In the BAV case, the axial velocity profile also evolves, but with greater asymmetry, featuring a slope towards the convex side (in the direction of the raphe between the left and right coronary leaflets). The velocity gradient is larger on this side compared to the TAV case, resulting in increased axial wall shear stress (Figure 7d). However, on the opposite side (x/X=1), there is a small gradient with low axial velocity, leading to a lower axial wall shear stress compared to the TAV case. This is also supported in the statistical comparison of the 3 TAV and 3 BAV cases (Figure 8).

The cross-flow velocity component (y-axis) in the BAV case exhibits more significant levels compared to the TAV case, as illustrated in Figure 7e,f, as well as in the statistical assessment in Figure 8b. This observation aligns with the presence of a strong swirl and higher axial vorticity, as shown earlier in Figure 6. Consequently, the steeper gradient in the cross-flow velocity manifests as elevated tangential wall shear stress on the endothelium, particularly near the center of the vortex at x/X=0.7.

In the BAV cases, the axial velocity changes signify transitioning from positive to negative flow, which coincides with the location of the vortex core at x/X=0.7. This change in direction results in a non-zero oscillatory shear index (OSI) in the axial flow direction. A similar behavior is observed in the TAV case, although the magnitude of flow reversal is lower. On the other hand, when considering the cross-flow velocity component, it consistently maintains a positive sign, indicating that the oscillatory shear index (OSI) in the tangential flow direction is close to zero in both cases.

## 4. Discussion

In this study, we employed machine learning (ML)-based segmentation techniques on an MRI dataset to segment the anatomical structures of the cardiovascular system. The ML-based segmentation approach, trained on CT datasets, exhibited a high level of accuracy, with a Dice score of 0.86 of the main chambers of the heart, which is in good agreement with segmentation using CT data [18]. However, its performance was relatively poorer when segmenting the aorta, achieving a Dice score of 0.72 in that particular region, which agrees with other studies that report better performance with CT compared to MRI [17]. Therefore, using MRI datasets on ML-based segmentation that is pre-trained on CT could affect the Dice score. In addition, the leaflet thickness of the aortic valve and the diameter of the coronary arteries are two features with dimensions below the resolution of 4D-PCMRI, which may also explain the segmentation failure and lower Dice score of the aorta.

The ML-based segmentation method demonstrated robustness when applied to clinical MRI data, effectively reducing the time required for the entire heart segmentation compared to manual segmentation. The runtime for the segmentation process was less than approximately 2 min, and it demanded about 5 GB of RAM and GPU memory. As a result, this approach can be executed on a workstation, making it practical and feasible for routine usage.

The functionally bicuspid aortic valve and trisinuate aortic root discussed in this study are more frequently linked to aortic valve narrowing and aortic enlargement [35]. In this latter type, two functional commissures exist with two normal interleaflet triangles, extending to the sinutubular junction. On the contrary, the interleaflet triangle below the fusion zone is underdeveloped without a functional commissure, as its tip does not reach the sinutubular junction. These data support the notion that fusion between the right and left leaflets leads to an asymmetric wall shear stress distribution, with increased circumferential wall shear stress compared to TAVs [36]. This can potentially lead to leaflet thickening, sclerosis, and eventual calcification, influencing aortic valve narrowing progression. Additionally, the study highlights that the BAV aortopathy directly impacts hemodynamics in the thoracic aorta, shedding light on potential risk factors for aortic enlargement progression [37].

These findings have important implications for assessing congenitally malformed valves and providing insights for surgical interventions. Studies have shown that the fusion orientation and the level of hypoplasty of the commissural apex can guide decisions between preserving a functional bicuspid valve or re-configuring it to a tricuspid valve, optimizing the potential for long-lasting repairs [1,38].

## 5. Limitation

The sub-optimal Dice score obtained for the aorta resulted in segmentation failure of the head and neck vessels, coronary arteries, and the fine details of the valve leaflets. In cases where segmentation failure occurs, it may be necessary to resort to manual segmentation, although this would incur additional turnaround time for the complete segmentation of the heart.

One limitation of the ML-based segmentation method employed in this study is that it was trained on ECG-gated CT datasets. Consequently, the study was constrained to utilizing 3D mDixon data that were ECG-gated. Therefore, the physiological displacement of the heart throughout the cardiac cycle could impact the delineation of the 4D-PCMRI data, which evolve both spatially and temporally. This influences the flow quantification in the left ventricle, as it experiences significant volume changes during systolic contraction. However, for vessels that undergo moderate displacement and area changes, this factor is less significant. Nonetheless, including 4D-PCMRI data in the training datasets for the ML-based segmentation would help overcome this limitation and enhance the segmentation performance. This will be the objective of a future study.

The study’s scope is constrained by the limited number of subjects, focusing on the functionally bicuspid aortic valve (BAV) with a trisinuate aortic root when compared to the normal trisinuate aortic valve (TAV). Nonetheless, the cohort size in this study is consistent with other studies [39]. Subsequent studies will employ this methodology for more intricate analysis of flow structures, incorporating a more extensive cohort that matches age, sex, and valve anatomy. This expanded dataset will include normal aortic valves as well as various forms of bicuspid and unicuspid aortic valves. These future investigations will aim to establish connections between these observations and the likelihood of aortic dilation [40].

## 6. Conclusions

The examined ML-based segmentation method demonstrated precise measurement of left ventricular volume and accurate identification of the thoracic aorta region, regardless of aortic pathology. These findings suggest that this method holds significant promise as a valuable tool for promptly assessing aortic pathologies like the bicuspid aortic valve in a clinical setting.

The quantification of flow structures in TAV and BAV morphologies was accomplished using ML-based segmentation to delineate the 4D-PCMRI flow measurements. Comparative analysis revealed that BAVs exhibited a more pronounced impingement of the jet towards the convex side of the ascending aorta, in contrast to TAVs. At peak systole, the axial vorticity data in the ascending aorta demonstrated that BAV subjects displayed a significant vortex structure with a counterclockwise swirl, which was more prominent than in the normal TAV cases. This is in agreement with previous observations [4,8,16,36]. This vortex structure led to increased tangential wall shear stress in BAV compared to TAV, concluding the significance of quantifying the tangential component of wall shear stress in addition to the axial wall shear stress.

The quantitative assessment of vortical flow structures contributes to a deeper understanding of the relationship between hemodynamics in the proximal thoracic aorta and differences in aortic valvar morphology. The Computational Fluid Dynamics model in ideal [41,42] and patient-specific [43,44,45] geometries will be implemented to compare with the observed behavior. Further investigations involving larger cohort sizes will be conducted to determine the potential clinical utility of these findings, particularly concerning the propensity for aortic dilation in individuals with BAV. However, it is anticipated that the main findings will remain consistent even with a larger cohort.

## Figures and Tables

**Figure 1 bioengineering-10-01216-f001:**
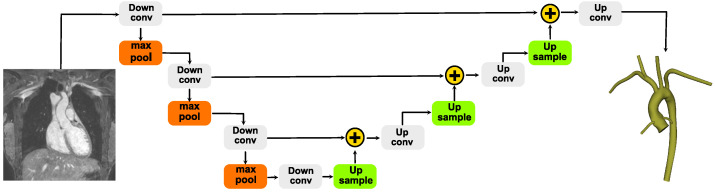
Schematics of a U-Net architecture. The raw MRI data are first down-sampled via convolution and max pool operations. The resulting latent space is up-sampled again via convolution and concatenation (yellow plus signs in the figure). The output is the segmented 3D domain.

**Figure 2 bioengineering-10-01216-f002:**
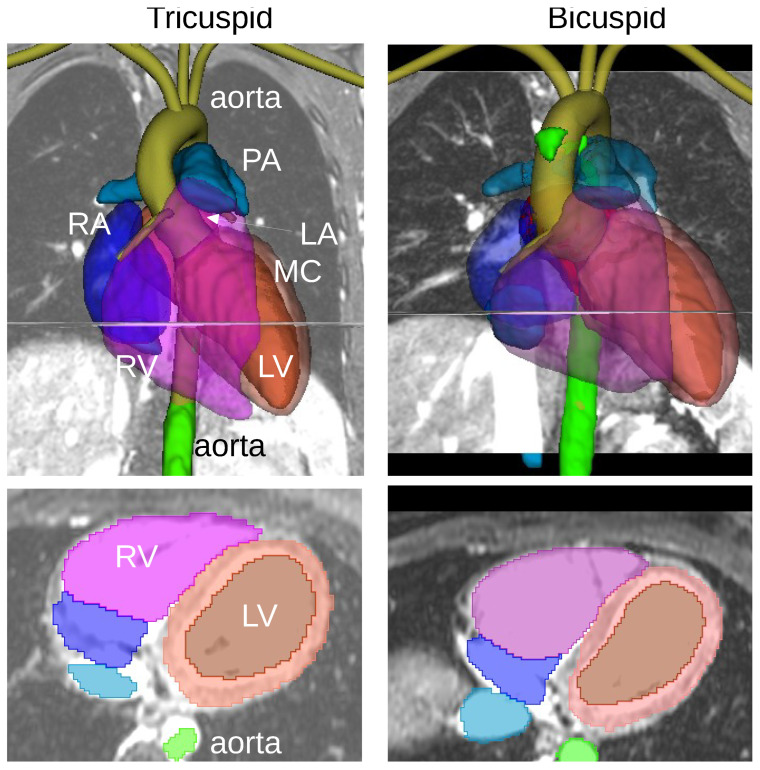
Segmentation of cardiovascular structures of the TAV (**top left panel**) and BAV (**top right panel**) subject by the ML-based segmentation. The segmented structures on the top coronal view are: the left ventricle (LV), myocardium (MC), left atrium (LA), right ventricle (RV), right atrium (RA), pulmonary artery (PA), aorta. The axial plane at the mid-position of the LV is shown in the (**bottom panels**).

**Figure 3 bioengineering-10-01216-f003:**
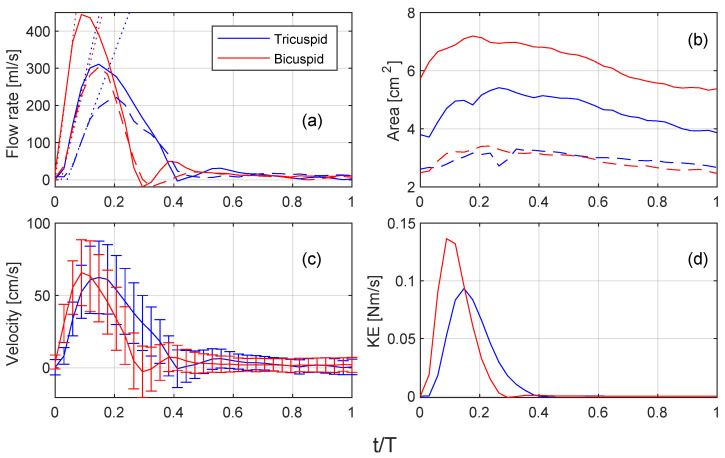
Comparison of different flow quantities for the TAV (blue) and BAV (red) cases: (**a**) ascending (full line) and descending (dashed line) flow, (**b**) area, (**c**) velocity, (**d**) kinetic energy. The dotted lines in (**a**) indicate the up slopes of the flow curves that are used for determining the time interval of the PWV.

**Figure 4 bioengineering-10-01216-f004:**
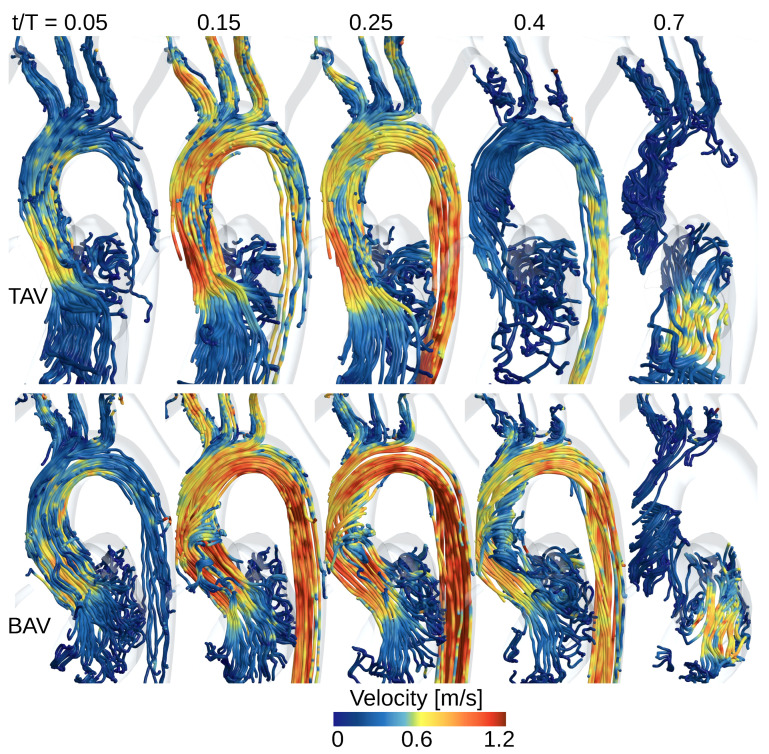
Streamlines colored by the velocity magnitude at different instances during the cardiac cycle for the TAV (**top row**) and BAV (**bottom row**) subjects. The streamlines show flow structures in the aorta, left atrium, and the left ventricle.

**Figure 5 bioengineering-10-01216-f005:**
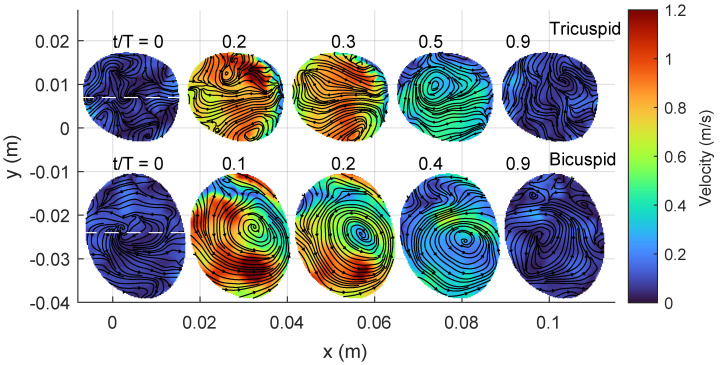
Short-axis view showing velocity magnitude with streamlines at the location proximally of the sinutubular junction in the ascending aorta (c.f. Figure 4 TAV (top row) and BAV (bottom row).

**Figure 6 bioengineering-10-01216-f006:**
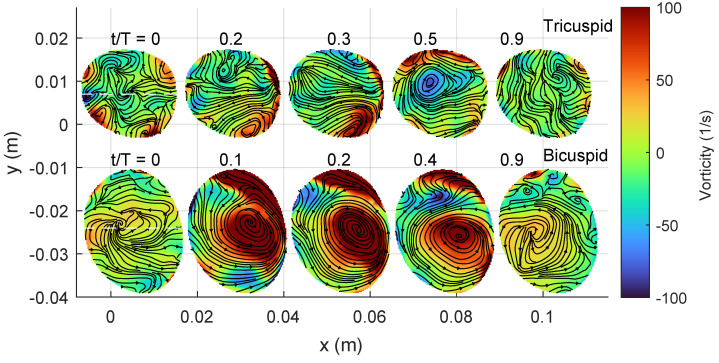
Short-axis view showing the axial vorticity with streamlines. Same keys as Figure 5.

**Figure 7 bioengineering-10-01216-f007:**
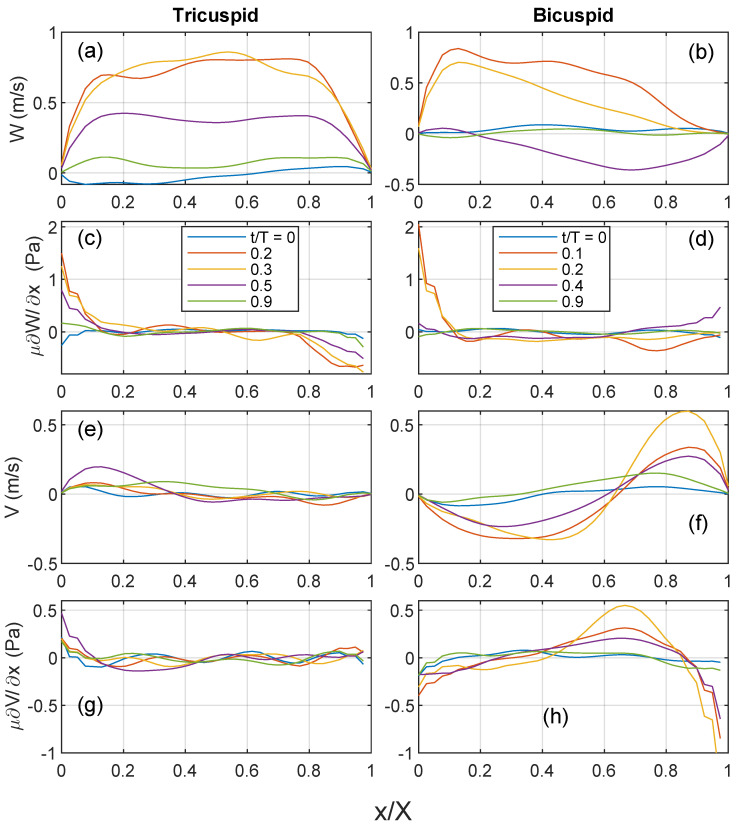
Quantification of velocity and shear stress along the dashed line annotated in Figure 5 that is located proximally of the sinutubular. The x-axis of the profile is normalized with the total length where x/X=0 is towards the convex side and x/X=1 is towards the concave side of the ascending aorta. (**a**,**b**) Streamwise velocity, (**c**,**d**) streamwise shear stress, (**e**,**f**) cross-flow velocity, (**g**,**h**) shear-stress in cross-flow direction. TAV (left column) and BAV (right column).

**Figure 8 bioengineering-10-01216-f008:**
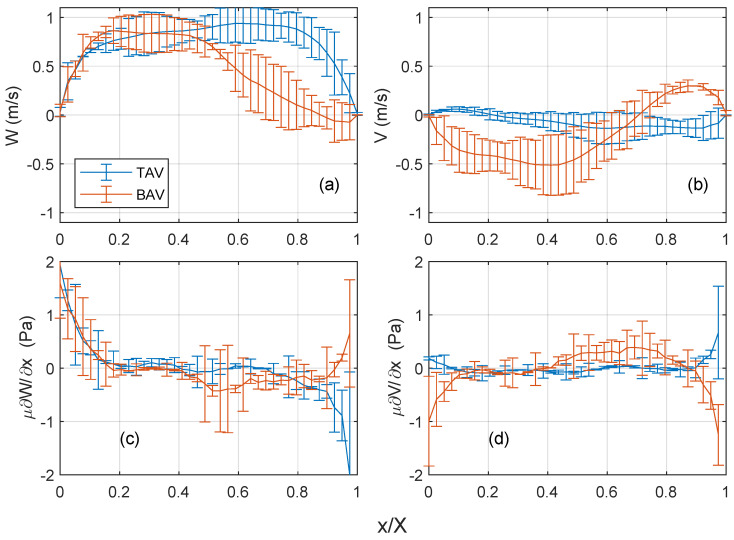
Error bar showing the mean and standard deviation of the velocity and shear stress, along the dashed line annotated in Figure 5 for all considered cases, i.e., 3 TAV and 3 BAV cases. The same keys as in Figure 7, but here the data are during peak systole.

**Table 1 bioengineering-10-01216-t001:** Segmented cross-sectional areas of the left ventricle and the aorta measured on the axial cut shown in Figure 2. The ground truth values (in parentheses) of the left ventricle and the aorta were obtained by manually tracing a closed loop around the region of interest. The mean μ and standard deviation σ of the Dice score (in square brackets) are given for both TAV and BAV cases.

Parameter	Aorta (cm^2^)	Left Ventricle (cm^2^)
TAV1	1.4 (2.0) [0.82]	17.4 (20.9) [0.91]
TAV2	1.3 (1.9) [0.81]	11.1 (14.2) [0.88]
TAV3	0.9 (2.1) [0.60]	18.4 (22.5) [0.90]
μ and σ of Dice score	[0.72 ± 0.12]	[0.86 ± 0.06]
BAV1	2.9 (2.2) [0.86]	24.1 (27.1) [0.94]
BAV2	2.2 (3.5) [0.75]	20.8 (29.3) [0.83]
BAV3	2.7 (3.7) [0.84]	14.3 (17.8) [0.89]
μ and σ of Dice score	[0.82 ± 0.06]	[0.89 ± 0.06]

**Table 2 bioengineering-10-01216-t002:** Image size, runtime, RAM, and GPU memory requirements of the TAV and BAV cases.

Case	Size (Voxels) (mm)	Runtime	RAM	GPU Mem
TAV	(320 × 320 × 100) (0.9 × 0.9 × 2.4 mm)	1 min 48 s	5.1 GB	3.0 GB
BAV	(400 × 400 × 100) (0.8 × 0.8 × 2.8 mm)	2 min 2 s	5.4 GB	3.2 GB

**Table 3 bioengineering-10-01216-t003:** Different parameters of the cardiac output calculated from the aortic ascending and descending flow.

Parameter	TAV	BAV
Heart rate (bpm)	70	47
Net volume (mL)	67	108
Ascending flow (L/min)	4.7	5.1
Regurgitant fraction (%)	0.3	1.5
Descending flow (L/min)	3.4	3.2
Aortic length (mm)	112	156
PWV time to foot (m/s)	3.9	3.8
Distensibility (1/mmHg)	0.008	0.009
ΔA/A	0.3	0.27
ΔP (mmHg)	36	31

## Data Availability

Data available on request due to restrictions in repository access.

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
