# Peer review of "Machine Learning-Based Segmentation of the Thoracic Aorta with Congenital Valve Disease Using MRI"

_bioengineering, 2023, doi:10.3390/bioengineering10101216_

Round 1
Reviewer 1 Report
This study showed that the machine learning-based segmentation reduces the runtime of the cardiac assessment and provides insights into vortical structures and the distribution of wall shear stress that may increase the risk of valve incompetence due to congenital aortic valvar disease. However, some problems need to be raised, and the writing and thinking of this document are very confusing.
1. Why are there many "question marks" in the reference notes in the article, which makes people feel that the writing is very imprecise?
2. In the method part, the information on research subjects, number of people, age, sex and so on is not provided.
3. What are the magnetic resonance sequence parameters? What is the magnetic resonance instrument's specific information (brand, country)? None of this essential information was provided.
4. It is puzzling that there is no statistical analysis in the result part.
5. The discussion part is too simple to analyze the significance and role of machine learning in this research.
6. The conclusion is unclear, and the argument is not prominent.
The language expression is not clear and accurate enough.
Reviewer 2 Report
Overall, this paper presents a promising approach to improve cardiovascular assessment in subjects with congenital valve disease, specifically bicuspid aortic valves (BAV), by employing machine learning-based segmentation techniques on MRI data. The authors have made significant progress in their research; however, several major revisions are necessary to enhance the quality and clarity of the paper.
Abstract Clarity: The abstract provides a concise overview of the study but lacks clarity in some areas. The objectives and significance of the research should be more clearly defined, and the abstract should briefly mention the key findings.
Methodology Detail: The paper mentions that the machine learning-based segmentation model was trained on whole-body CT, but it lacks specific details about the model architecture, training dataset, and hyperparameters used. Provide more technical information to enable reproducibility.
Results Presentation: The results are generally well-presented, but the Dice scores for both the heart organ and the thoracic aorta should be discussed in more detail. Explain why the thoracic aorta segmentation achieved a lower Dice score compared to the heart organ. Provide insights into potential challenges or limitations of the segmentation model.
Discussion Expansion: The discussion section should be expanded to provide a deeper analysis of the implications of the findings. Specifically, explain how the observed asymmetry in wall shear stress in BAV cases may impact clinical outcomes or patient management. Connect the results to the broader context of cardiovascular assessment.
Statistical Analysis: If applicable, include statistical tests or analyses to support the significance of the observed differences in wall shear stress between TAV and BAV cases. This will strengthen the paper's scientific rigor.
Conclusion Clarity: The conclusion should be rewritten to summarize the main findings and their implications more concisely. Highlight the practical applications of ML-based segmentation in clinical settings for patients with congenital aortic valve disease.
Citation and References: Ensure that all sources are properly cited throughout the paper, and verify the accuracy of the references in the bibliography. Reference numbers are missing, e.g., line 80 278 etc.
Incorporating these major revisions will significantly enhance the quality and impact of your paper, making it a valuable contribution to the field of cardiovascular assessment in congenital valve disease using machine learning-based segmentation techniques.
Easy to follow up, but can be further improved.
Reviewer 3 Report
This is a well done study on the very up-to-date clinical issue. Machine learning and artificial intelligence can facilitate assessment and interpretation of time-consuming examinations, including the diagnosis of the thoracic aorta variants/anomalies and pathologies. The Authors have used two modalities: cardiac MRI data sets, which included 3D mDixon angiogram and 4D phase-contrast magnetic resonance imaging (4D-PCMRI) for the assessment of aortic valve BAV and TAV. They have used CT as point of reference.
The imaging studies of aortic valve and thoracic aorta require much experience and they are time consuming. Thus, the machine learning, particularly for thoracic aorta, are worth of exploring. Although, the Authors compare MRI machine learning in relation to BAV vs TAV, I believe they could explore this issue wider, at least in Introduction and Discussion. Particlularly, Discussion is very short. I think it could address some issues, including risk of the aorta and aortic arch variants and anomalies (like bovine arch, variations in course and origin of aortic arch side-branches, risk of rupture or dissection which is higher in the presence of BAV, etc.). Here, in patients with BAV, the shear stress and collagen tissue composition in the aorta are very important ( I think the Authors may find utile to address this issue: https://doi.org/10.3390/jcm12123949 )
Thus, my major comments are:
Introduction. It seems that the main issue in the study in BAV. But there are many other thoracic aorta pathologies that require emergent and accurate diagnosis. I believe that the paper would benefit from the boarder presentation of thoracic aorta pathologies. The Authors could find utile the following study: https://doi.org/10.3390/biomedicines11082207 . It decribes emergency clinical scenarios in which CTA and MRA of thoracic aorta are the first line diagnostic strategy, and BAV makes important contribution. In line with this, a machne learning would facilitate the diagnosis and therapeutic approach (https://doi.org/10.3390/diagnostics12081790)
In line with the Authors' observation, also the other researchers found better performance of CT compared to MR
Minor comments:
Please develop the abbreviation 'TAV' in abstract
Introduction, Methods: please remove question tags: [2? ? ? ], and verify the reference
Conclusions: Computational Fluid Dynamics model in ideal [31? ] and patient-specific [? ? ? ]
Round 2
Reviewer 2 Report
The authors have addressed the review comments.
Suggest to carefully proof read this manuscript again.
Reviewer 3 Report
The Authors answered all comments in satisfactory way. I have no further comments.